# Immunoinformatics and Evaluation of Peptide Vaccines Derived from Global Hepatitis B Viral HBx and HBc Proteins Critical for Covalently Closed Circular DNA Integrity

**DOI:** 10.3390/microorganisms11122826

**Published:** 2023-11-21

**Authors:** Umar Saeed, Zahra Zahid Piracha, Salman Alrokayan, Tajamul Hussain, Fahad N. Almajhdi, Yasir Waheed

**Affiliations:** 1Clinical and Biomedical Research Center (CBRC) and Multidisciplinary Laboratory (MDL), Foundation University Islamabad, Islamabad 44000, Pakistan; umarsaeed15@yahoo.com; 2Department of Microbiology, Ajou University School of Medicine, Suwon 443-749, Republic of Korea; piracha.zahra@gmail.com; 3International Center of Medical Sciences Research (ICMSR), Islamabad 44000, Pakistan; 4Research Chair for Biomedical Application of Nanomaterials, Biochemistry Department, College of Sciences, King Saud University, Riyadh 11362, Saudi Arabia; salrokayan@ksu.edu.sa; 5Center of Excellence in Biotechnology Research, College of Applied Medical Sciences, King Saud University, P.O. Box 10219, Riyadh 11451, Saudi Arabia; 6Botany and Microbiology Department, College of Science, King Saud University, P.O. Box 2455, Riyadh 11451, Saudi Arabia; majhdi@ksu.edu.sa; 7Office of Research, Innovation, and Commercialization, Shaheed Zulfiqar Ali Bhutto Medical University, Islamabad 44000, Pakistan; yasir_waheed_199@hotmail.com; 8Gilbert and Rose-Marie Chagoury School of Medicine, Lebanese American University, Byblos 1401, Lebanon

**Keywords:** global HBV, HBx, HBc, cccDNA, peptide vaccines

## Abstract

The Hepatitis B virus (HBV) HBx and HBc proteins play a crucial role in associating with covalently closed circular DNA (cccDNA), the primary factor contributing to intrahepatic viral persistence and a major obstacle in achieving a cure for HBV. The cccDNA serves as a reservoir for viral persistence. Targeting the viral HBc and HBx proteins’ interaction with cccDNA could potentially limit HBV replication. In this study, we present epitopes identified from global consensus sequences of HBx and HBc proteins that have the potential to serve as targets for the development of effective vaccine candidates. Furthermore, conserved residues identified through this analysis can be utilized in designing novel, site-specific anti-HBV agents capable of targeting all major genotypes of HBV. Our approach involved designing global consensus sequences for HBx and HBc proteins, enabling the analysis of variable regions and highly conserved motifs. These identified motifs and regions offer potent sites for the development of peptide vaccines, the design of site-specific RNA interference, and the creation of anti-HBV inhibitors. The epitopes derived from global consensus sequences of HBx and HBc proteins emerge as promising targets for the development of effective vaccine candidates. Additionally, the conserved residues identified provide valuable insights for the development of innovative, site-specific anti-HBV agents capable of targeting all major genotypes of HBV from A to J.

## 1. Introduction

The Hepatitis B virus (HBV), a member of the Hepadnaviridae family, exhibits exclusive tropism for hepatocytes and possesses small genomes comprising partially double-stranded, partially single-stranded circular DNA (pdsDNA) [1]. HBV infection initiates acute Hepatitis, progressing to chronic Hepatitis B (CHB), liver fibrosis, fulminant hepatic failure, cirrhosis, and hepatocellular carcinoma (HCC) [1]. Globally, 257 million individuals live with CHB, resulting in 887,000 deaths annually [2].

HBV encompasses 10 major genotypes (A, B, C, D, E, F, G, H, I, and J), each with diverse geographical distributions [3]. Despite the existence of anti-HBV vaccines, viral Hepatitis remains a significant public health challenge [2]. Commercially available reverse transcriptase inhibitors, including tenofovir, lamivudine, and entecavir, offer treatment for HBV patients, but necessitate lifelong administration [4]. Presently, no cure for HBV exists [5]. A genuine cure for HBV requires the clearance of intra-hepatic nuclear covalently closed circular DNA (cccDNA), a pivotal factor in virological persistence [5,6]. Throughout the HBV life cycle, cccDNA associates with cellular histones, non-histone proteins, including HBc and HBx, transcription factors, co-activators, and various epigenetic activators and repressors, influencing HBV transcription and epigenetic control [7,8,9].

HBx, a 154-amino acid protein, stimulates viral replication significantly, playing a vital role in initiating and sustaining the HBV life cycle, fostering host–virus interactions, and contributing to HCC development [9,10]. In the absence of HBx, cccDNA rapidly adopts a silent state (closed confirmation), becoming transcriptionally inactive [6]. Silencing the HBx gene through short interfering RNA (siRNA) inhibits HBV gene expression and replication in HBV genotype B hydrodynamic injection mouse models [11]. A recent study demonstrated that cell-penetrating antibodies targeting HBx in chronic HBV infection mimicking mouse models or cells substantially suppressed the Hepatitis B virus [12].

HBc, a protein of 183 or 185 amino acids (length dependent on strains), facilitates the formation of the nuclear capsid capable of incorporating pre-genomic (pgRNA) and reverse transcriptase. It drives the conversion of single-stranded (SS) DNA to relaxed circular (RC) DNA, generating mature nucleocapsids, which can be enveloped through envelope proteins and released extracellularly as infectious virions [13,14,15].

HBV covalently closed circular DNA (cccDNA) associates with cellular and viral DNA binding proteins, HBc (via direct binding), and HBx (through indirect binding), forming a viral episome serving as a template for viral transcription and protein expression within HBV-infected hepatocyte nuclei [8,9]. HBx promotes a transcriptionally active state of the viral episome, enhancing HBV replication significantly, while HBc plays crucial roles in the structural organization of the episome, altering nucleosome numbers and leading to higher episome copy numbers, as depicted in Figure 1. Targeting cccDNA-binding viral proteins (HBc and HBx) emerges as a crucial antiviral approach to limit HBV replication.

Recently developed HBV vaccines have proven effective in preventing Hepatitis B. However, existing anti-HBV therapeutic options fall short of eradicating the disease, as a functional cure remains elusive. Chronic HBV is associated with dysfunctional B or T cells, and while HBV-specific T-cell dysfunction is not irreversible, it holds the potential for restoration through anti-HBV vaccination. Presently, available HBV vaccine candidates, when administered to individuals with chronic HBV, do not appear to sufficiently enhance extensive HBV-specific T-cell responses. Potential strategies to enhance therapeutic vaccination for chronic Hepatitis B virus (HBV) infection may involve reducing antigenic load before vaccination, employing checkpoint inhibitors to reverse T-cell dysfunction, combining vaccine platforms in heterologous prime-boost strategies, targeting T cells to the liver, and optimizing immunogen sequences of multiple HBV antigens for reactivity across various genotypes.

Given the critical roles of both HBx and HBc in HBV replication, targeting cccDNA-binding viral HBc and HBx proteins emerges as a potential strategy to limit viral replication. Our objective was to design global consensus sequences of HBx and HBc proteins, analyzing variable regions and highly conserved motifs that could serve as potent target sites for the development of peptide vaccines, site-specific RNA interference, and anti-HBV inhibitors. Additionally, we explored B- or T-cell epitopes in HBx and HBc proteins using in silico approaches, aiming to identify epitopes capable of generating neutralizing antibodies against all A–J genotypes of HBV.

## 2. Material and Methods

### 2.1. Retrieval of HBx and HBc Sequences and Development of Global Consensus Sequences

A total of 237 HBx and 207 HBc sequences belonging to all major 10 genotypes of HBV were randomly retrieved from the National Center for Biotechnology Information database (https://www.ncbi.nlm.nih.gov/nuccore/ (accessed on 2 February 2023)). These sequences were reported from all over the world including the USA, Canada, Brazil, Venezuela, Argentina, France, Germany, The Netherlands, Mexico, Belgium, China, South Korea, Japan, Indonesia, Thailand, Viet Nam, Turkey, Pakistan, India, South Africa, Ghana, Côte d’Ivoire, Nicaragua, Mauritius, and Martinique. The HBx and HBc amino acid sequences were fed into CLC Main Workbench v. 8 (Qiagen GmbH, Hilden, Germany). Using the multiple sequence analysis feature of the software, we constructed consensus sequences of each genotype (A–J). The consensus sequences of all ten genotypes thus constructed were subsequently aligned to obtain global consensus sequences of HBx and HBc. 

### 2.2. Peptide Designing and Phylogenetic Analysis 

The global consensus sequences were analyzed to find highly conserved regions and variable residues in different domains and motifs of HBx and HBc proteins. Short stretches of amino acids from the highly conserved regions of HBx and HBc proteins were selected from the global consensus sequence alignments; these peptides could be better targets for potential peptide-based vaccines testing and useful for designing inhibitory compounds. To draw phylogenetic trees, all 237 HBx and 207 HBc sequences were aligned in CLC Main Workbench v. 8 (Qiagen GmbH, Hilden, Germany) and subjected to an unweighted pair group method with an arithmetic mean (UPGMA) with a bootstrap value of 100. 

### 2.3. B- and T-Lymphocyte Epitopes Prediction

The location of B- and T-cell epitopes was mapped in the global consensus sequences of HBx and HBc. Target epitopes for B-lymphocytes in HBx and HBc were predicted for possible antibody binding through the Kolaskarand Tongaonkar antigenicity prediction tool or method available via Immune Epitope Database (IEDB) (https://www.iedb.org/ (accessed on 2 February 2023)). The aforementioned method can predict antigenic determinants with about accuracy of 75%, better than most of the known methods. Possible T-lymphocyte epitopes in HBx and HBc were analyzed for promiscuous major histocompatibility complex class-I (MHC-I) and class-II (MHC-II) through ProPred-I (http://crdd.osdd.net/raghava/propred1/ (accessed on 15 February 2023)) and ProPred in silico prediction tools (http://crdd.osdd.net/raghava/propred/ (accessed on 20 February 2023)), respectively. Predicted B-cell and T-cell epitopes in HBx and HBc were further analyzed for epitope conservancy analysis via the IEDB analysis resource (http://tools.iedb.org/conservancy/ (accessed on 20 February 2023)). To analyze if these peptides could trigger autoimmunity, selected epitopes with 70–100% conservancy were compared with human proteome via Mimicry Peptide Database(miPepBase) (http://proteininformatics.org/mkumar/mipepbase/ (accessed on 22 February 2023)). 

## 3. Results

### 3.1. The Global Consensus Sequence of HBx and HBc and Analysis of Highly Conserved Regions

We hypothesized that conserved residues in HBx and HBc are crucial for developing novel anti-HBV agents, designing peptide-based vaccines, and site-specific inhibitors. Therefore, to analyze highly conserved domains of HBx and HBc, we constructed consensus sequences for all ten major global genotypes of HBV and aligned the genotypic consensus sequences to develop global consensus sequences for HBx and HBc, respectively.

Figure 2 illustrates the alignment of consensus sequences of HBx from all ten HBV (A–J) genotypes, with the global consensus sequence presented at the base. Various motifs and domains of HBx were scrutinized for amino acid conservancy and variability (Figure 2A). The graphs depict the percentage conservancy of HBx amino acids, with conserved residues labeled by their symbols, while variable amino acids are denoted by the symbol “X.” Regions with highly conserved amino acids could potentially serve as sources for peptide vaccines [16,17]. Therefore, small peptide fragments (Table 1) were derived from the highly conserved regions of the HBx global consensus sequence (Figure 2A,B).

Additionally, a phylogenetic tree was constructed (Figure 3) using 237 HBx sequences representing genotypes A–J, reported from various countries worldwide. The sequences from different genotypes clustered together based on evolutionary relatedness.

Similarly, we aligned HBc consensus sequences from all ten HBV genotypes, constructed a global consensus sequence, shown at the base (Figure 4A), and analyzed various domains and motifs for conservancy or variability of amino acids. The graphs depict the percentage conservancy of HBc amino acids. Highly conserved residues are shown by their symbols, while variable regions are labeled by “X”. Short peptides were selected (Table 2) from highly conserved regions of the HBc global consensus sequence (Figure 4A,B), which might offer potent target sites for the development of a peptide vaccine or designing site-specific anti-HBV inhibitors. The phylogenetic tree of 207 HBc sequences from 10 genotypes worldwide was constructed (Figure 5), indicating evolutionary relatedness between the sequences.

### 3.2. B-Lymphocyte Epitopes of HBx and HBc Proteins and Conservation Analysis

Global consensus sequences of HBx and HBc proteins were subjected to analysis in IEDB for the prediction of different B-cell epitopes which might produce neutralizing antibodies against all HBV genotypes. Each HBx-related B-cell epitope was given a distinctive name, from X-B1 to X-B7 (Table 3), while HBc-related B-cell epitopes were denoted as C-B1 to C-B7 (Table 4). The predicted epitopes were subjected to epitope conservation analysis via an IEDB conservation analysis resource. The location, length, and percentage conservancy of each epitope of HBx and HBc are given in Table 3 and Table 4, respectively. Among HBx-related B-cell epitopes, X-B2 (DVLCLRP) and X-B4 (AGPCALR) were considered to be conserved in all ten genotypes (Table 3). However, among HBc-related B-cell epitopes, C-B6 (IRQLLWFHISCLTF) and C-B7 (VLEYLVSFGV) were considered highly conserved in all HBV genotypes (Table 4). We utilized the miPep Base facility to analyze whether these peptides trigger autoimmunity. And none of the peptides triggered an autoimmune response. The aforementioned epitopes, predicted from global consensus sequences, might be capable of generating strong neutralizing antibodies against all (A–J) genotypes of HBV.

### 3.3. T-Lymphocyte MHC-I- and MHC-II-Specific Epitopes of HBx and HBc Proteins and Conservation Analysis

The location of possible T-cell MHC-I epitopes was identified in the global consensus sequence of HBx and HBc through ProPred-I, for 47 MHC-I alleles. In total, 323 different HBx-related MHC-I epitopes were predicted, which were subsequently subjected to conservation analysis via IEDB epitope conservation analysis. We selected epitopes with 70–100% conservancy (Table 5). Among these epitopes, X-M2, X-M5, X-M8, X-M11, X-M12, X-M20, X-M22, X-M25, X-M27, and X-M32 were conserved in the HBV HBx consensus sequence and among all ten genotypes of HBV. Similarly, 182 different HBc-related MHC-I epitopes were predicted and subjected to epitope conservation analysis. Epitopes with 70–100% conservancy were selected and presented in Table 6. The C-M1, C-M2, C-M4, C-M6-11, C-M13, C-M19, C-M24-26, C-M30, C-M34-36, C-M40, and C-M43-45 epitopes remained conserved in the HBc consensus sequence and all HBV genotypes.

To identify the possible location of T-cell MHC-II epitopes in the global consensus sequence of HBx and HBc, we utilized the Propred in silico analysis facility, for 51 HLA-DR alleles. In total, 204 HBx-related MHC-II epitopes were predicted, which were subjected to IEDB epitope conservancy analysis. We selected epitopes having 70–100% conservancy, as shown in Table 7. The X-T4, X-T6, and X-T8 epitopes were conserved in the HBx consensus sequence and all HBV genotypes. Similarly, 203 different HBc-related MHC-II epitopes were predicted and analyzed for epitope conservancy as described above. Epitopes with 70–100% conservancy are presented in Table 8. Among these, the C-T1-3, C-T9, C-T10, C-T12, C-T14, and C-T16-18 epitopes were identified to be 100% conserved in the HBc consensus sequence and among all (A–J) genotypes of HBV. Using miPepBase, we found that none of the peptides trigger an autoimmune response.

## 4. Discussion

The HBx protein comprises regulatory and transactivation domains, as illustrated in Figure 2B. The regulatory domain, spanning amino acids 1 to 50, is non-essential for HBx activity and even suppresses HBx transactivation, as reported previously [18]. A consensus sequence analysis of the regulatory domain reveals that the region from 1M to 20P is highly conserved, whereas the region from 31S to 40P varies, except for the 32G and 35G residues, which remain highly conserved across all HBV genotypes.

Conversely, the transactivation domain, covering amino acids 51 to 154, is indispensable for enhancing HBV transcription and replication [19]. Specifically, the amino acids within the 52H to 65S region play a critical role in augmenting HBV replication [19]. The consensus analysis shows that the HLSLR-GLPVCAFSS motif is highly conserved across all HBV genotypes. Removing the last 14 amino acids (141L to 154A) of the transactivation domain does not impact its transactivation properties [20]. It has been established that amino acids 132F to 140K, particularly 137C, are crucial for HBx transactivation [20]. The consensus sequence analysis highlights that the FVLGGCRHK motif and the 137C residue are fully conserved among all HBV genotypes. This suggests that designing anti-HBV siRNA or inhibitors targeting this region could effectively target HBx in all HBV genotypes. However, a natural HBx mutant, known as HBxDelta127, lacks the FVLGGCRHK motif and 137C, and has been observed in patients with chronic Hepatitis B, liver cirrhosis, and hepatocellular carcinoma (HCC). This mutant can induce the growth and proliferation of hepatoma cells [21,22].

Moreover, the presence of a BH3-like motif (110A to 135G) in the C-terminal region of the HBx protein allows it to directly interact with anti-apoptotic Bcl-2 family proteins and elevate cytosolic calcium levels, promoting viral DNA replication [23,24,25]. Consensus sequence analysis reveals that residues 111Y, 113K to 115C, 117F, 120W to 122E, and 132F to 135G within the BH3-like motif are completely conserved in all HBV genotypes. Among the 13 XaaP motifs in the HBx protein, the motifs 10D11P, 19R20P, 28R29P, and 45V46P in the regulatory domain, as well as 58L59P, 67G68P, and 89L90P in the transactivation domain, are entirely conserved across all HBV genotypes.

Moving on to the HBc protein, it consists of N-terminal (1 to 140 aa), linker (141 to 149 aa), and C-terminal (150 to 183 or 185 aa, depending on the strains) domains [26], as depicted in Figure 4B. The N-terminal domain (NTD) is crucial and sufficient for capsid assembly [27,28]. A consensus sequence analysis of the C-terminal domain (CTD) reveals that regions 1M to 11A are completely conserved among all HBV genotypes, except for genotype G, which contains an additional 12 amino acids RTTLPYGLFGLD insertion. This insertion has pleiotropic effects on core protein expression, HBV replication, and virion secretion [29]. In genotype G, the region 13V to 39R (or 25V to 51R of genotype G) remains highly conserved across all HBV genotypes. The NTD also carries a protease-like sequence, LLDTAS, which is reminiscent of retroviral proteases [30]. Consensus sequence analysis shows that this motif is highly conserved in all HBV genotypes. However, the region between 74 to 101 amino acids is considered hypervariable and could contribute to the development of liver injury [31,32]. Consensus sequence analysis also reveals that the 74X and 179X residues are the most variable among all HBV genotypes.

The linker domain, STLPETTVV, can interfere with NTD, pgRNA packaging in a sequence-independent manner, viral DNA synthesis during the first step of reverse transcription to initiate single-strand DNA, and in a sequence-dependent manner during the second step of reverse transcription for extensive plus-strand DNA synthesis to generate relaxed circular DNA. Additionally, it plays a role in virion secretion [26]. The presence of only five amino acids, ETTVV, in the linker region is sufficient to trigger single-stranded DNA synthesis [26]. Consensus sequence analysis indicates that the linker STLPETTVV region is entirely conserved among all HBV genotypes, except for genotype E, which contains STLPENTVV. The four cysteine residues at positions 48, 61, 107, and 183 are not essential for core particle formation, but they can further stabilize HBV core particles or HBc dimers [33,34]. The consensus sequence analysis shows that all of these cysteine residues are 100% conserved among all HBV genotypes. Amino acid regions between 98R to 115V and 117E to 145E are 100% conserved across all HBV genotypes, indicating the significance of this region in the HBV life cycle and/or viral pathogenesis.

The CTD contains highly basic residues (arginine rich, protamine like) resembling histone tails, which are crucial for non-specific nucleic acid binding [35,36]. The CTD is not required for capsid assembly but plays a significant role in pgRNA packaging and reverse transcription [37,38,39]. The phosphorylation of the CTD is essential for specific viral RNA packaging [40,41,42]. The three major phosphorylation sites in the CTD are 155S, 162S, and 170S, while there are four minor sites at 160T, 168S, 176S, and 178S [42,43,44,45]. The consensus sequence analysis demonstrates that, except for genotype H, which contains 155A instead of 155S, all the aforementioned phosphorylation sites in the CTD are 100% conserved across all HBV genotypes. The arginine-rich CTD domain contains distinct nuclear localization and cytoplasmic retention signals [46]. The primary RNA-recognizing activity of CTD is attributed to the 150R to 157R sequence [47]. The RRRR following 149V also contributes to RNA binding [35]. Consensus sequence analysis indicates that this RRRGRSPR motif is highly conserved among all HBV genotypes. In HBV genotype A, between amino acids 152R and 153G, two additional amino acids, DR, are present, generating the RRRDRGR-SPR motif. The DNA-recognizing activity of CTD is attributed to three repeated SPRRR motifs within the amino acid sequences from 157 to 177. Consensus sequence analysis reveals that these motifs are fully conserved among all HBV genotypes, except for genotype H, which contains 155A. Among the 16 XaaP motifs in the HBc protein, the motifs 4D5P, 19L20P, 24F25P, 78D79P, 128T129P, 129P130P, 133R134P, 134P135P, and 137A138P in NTD, 143L144P in the linker domain, and 160T161P, 162S163P, and 170S171P motifs in CTD are entirely conserved across all HBV genotypes.

Recent discoveries have underscored the active roles of HBc and HBx in the epigenetic regulation of the virus–host interplay [9,48]. The multifaceted functions of HBx and HBc, along with their influence on cccDNA mini-chromosome for enhanced viral replication, position these proteins as promising targets for antiviral therapeutics [9,48]. The consensus sequences of HBx and HBc were used to predict highly conserved B- and T-cell binding epitopes. Several highly or semi-conserved B-cell binding epitopes were identified, and we selected highly conserved epitopes with 70–100% conservation across all ten HBV genotypes. Similarly, numerous MHC-I or -II related epitopes demonstrated maximum allele-binding affinity, suggesting potential T-cell-related epitopes. Among the HBx-related B-cell binding epitopes, the X-B2 and X-B4 epitopes stand out due to their complete (100%) conservancy across all HBV genotypes, making them excellent targets for B-cell-based vaccine development. Likewise, among HBc-related B-cell binding epitopes, the C-B6 and C-B7 epitopes, with high (80%) conservancy across all HBV genotypes, are promising candidates for B-cell-based vaccine development. On the other hand, among HBx-related MHC-I-specific epitopes, X-M2, X-M5, X-M8, X-M11, X-M12, X-M20, X-M22, X-M25, X-M27, and X-M32, and among MHC-II-specific epitopes, X-T4, X-T6, and X-T8, could be considered for a synthetic vaccine against multiple genotypes of HBV. Similarly, for HBc-related MHC-I-specific epitopes, C-M1, C-M2, C-M4, C-M6-11, C-M13, C-M19, C-M24-26, C-M30, C-M34-36, C-M40, and C-M43-45, and among MHC-II-specific epitopes, C-T1-3, C-T9, C-T10, C-T12, C-T14, and C-T16-18, are ideal choices with high conservancy across all HBV genotypes. Using conserved epitopes predicted from global consensus sequences against NS3-4A could offer broader protection against multiple isotypes of the Hepatitis C virus [49]. Given the significant role of viral HBx and HBc proteins in association with cccDNA in promoting HBV replication, achieving a functional HBV cure remains a formidable challenge [50,51,52,53,54,55,56,57,58]. This study paves the way for designing innovative anti-HBV vaccines and therapeutics. Our study suggests that conserved epitopes from HBx and HBc global consensus sequences can be leveraged as potential targets for the development of effective vaccine candidates. Additionally, conserved residues can be harnessed for designing novel, site-specific anti-HBV agents capable of targeting all major HBV genotypes.

## 5. Conclusions

The Hepatitis B virus (HBV) proteins HBx and HBc play pivotal roles in augmenting viral replication, primarily by associating with covalently closed circular DNA (cccDNA)—a key factor in intrahepatic viral persistence and a major impediment to achieving a cure for HBV. The conservation of specific residues, such as 52H to 59P in HBx crucial for the augmentation effect in HBV replication, and 137C pivotal for transactivation, is observed across all HBV genotypes. Essential epitopes, including X-B2 and X-B4 for HBx, and C-B6 and C-B7 for HBc, are crucial for B-cell-based vaccines. Among the MHC-I and MHC-II epitopes for HBx, X-M2, X-M5, X-M8, X-M11, X-M12, X-M20, X-M22, X-M25, X-M27, and X-M32, and X-T4, X-T6, and X-T8, respectively, are vital for vaccine development. Crucial residues in HBc, specifically 141S to 149V, essential for pre-genomic RNA packaging, viral DNA synthesis, and virion secretion, exhibit high conservation across all HBV genotypes. Notably, MHC-I epitopes for HBc, including C-M1, C-M2, C-M4, C-M6-11, C-M13, C-M19, C-M24-26, C-M30, C-M34-36, C-M40, and C-M43-45, and MHC-II epitopes, such as C-T1-3, C-T9, C-T10, C-T12, C-T14, and C-T16-18, show significant conservancy and could be critical for peptide vaccines. Epitopes X-B2 and X-B4 for HBx and C-B6 and C-B7 for HBc emerge as noteworthy candidates for B-cell-based vaccines. The consensus sequence of HBx and HBc facilitates the screening of novel anti-HBV agents and the design of site-specific inhibitors with potential responsiveness against all HBV genotypes. Furthermore, predicted B- or T-cell epitopes hold promise for the development of antibodies effective against major HBV genotypes globally. While this study indicates B-cell or T-cell-related antigens based on in silico analysis, further characterization of the antigenic potential of these peptides in HBV infection animal models is warranted.

## Figures and Tables

**Figure 1 microorganisms-11-02826-f001:**
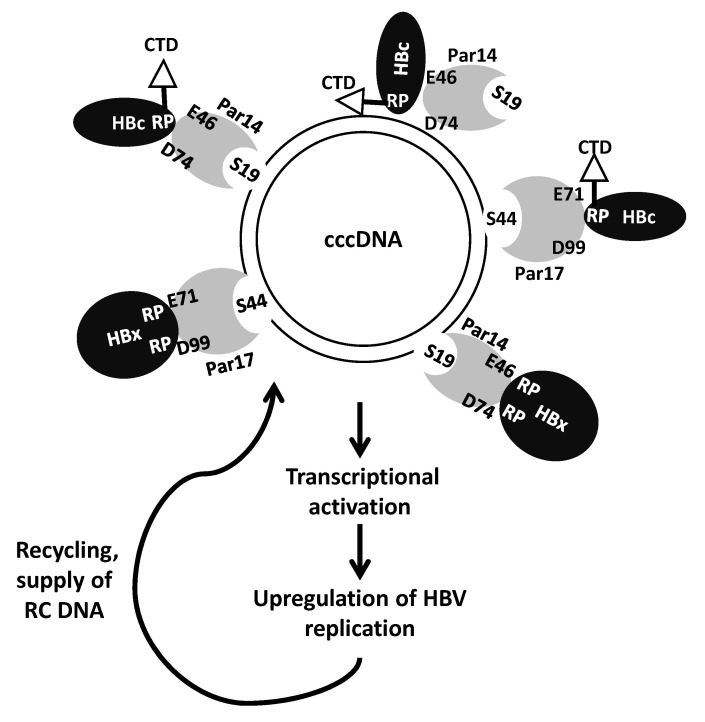
The HBV cccDNA minichromosome associates with histone and non-histone cellular and viral proteins, including HBc and HBx. In the nucleus, HBc binds to cccDNA and host proteins, such as Par14 and Par17, which can directly bind to cccDNA, interact with HBc, and promote the recruitment of HBc into cccDNA, consequently upregulating HBV replication by several folds.

**Figure 2 microorganisms-11-02826-f002:**
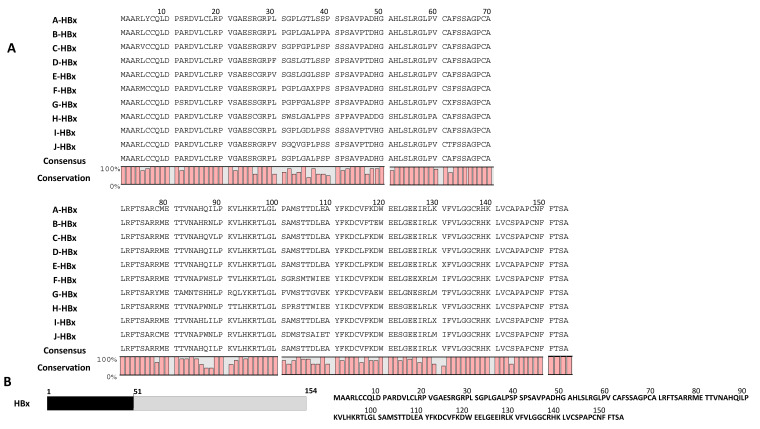
Sequence alignment of HBV genotype-specific consensus sequences of the HBx protein and global consensus sequence is shown. (**A**) Global consensus HBx sequence analysis among A to J genotypes of HBV. (**B**) Domain Structure and representative sequence of HBx protein.

**Figure 3 microorganisms-11-02826-f003:**
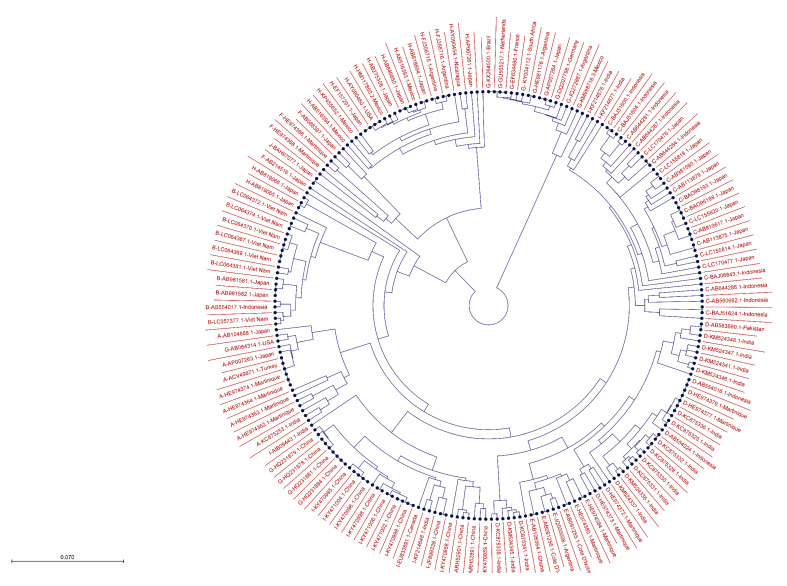
Phylogenetic tree of 237 HBV HBx sequences from all ten genotypes reported across the world.

**Figure 4 microorganisms-11-02826-f004:**
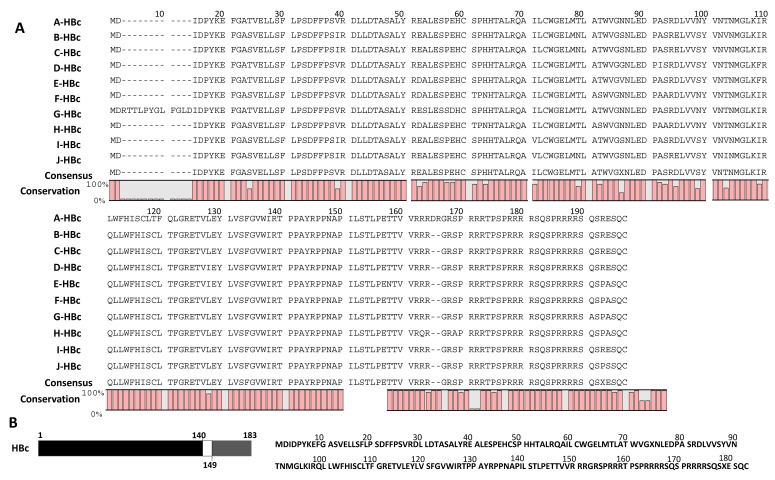
Sequence alignment of HBV genotype-specific consensus sequences of the HBx protein and global consensus sequence is shown. (**A**) Global consensus HBc sequence analysis among A to J genotypes of HBV. (**B**) Domain Structure and representative sequence of HBc protein.

**Figure 5 microorganisms-11-02826-f005:**
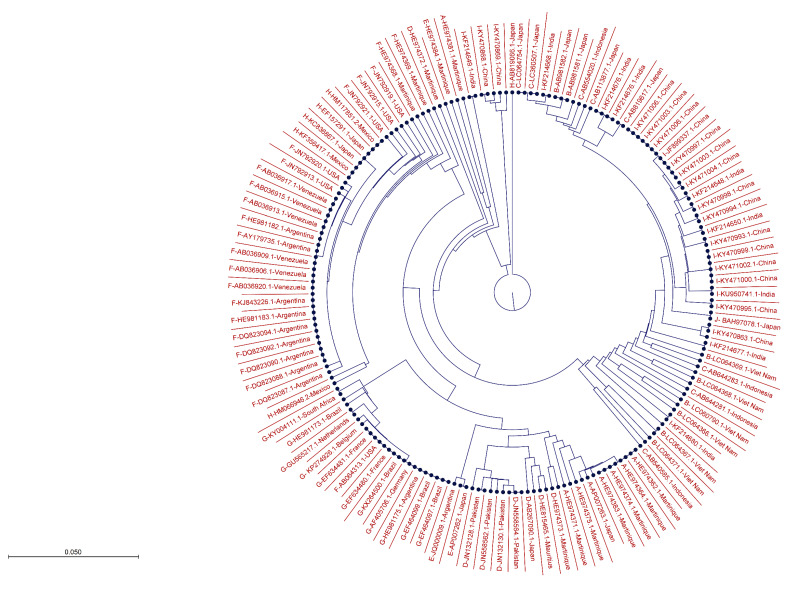
Phylogenetic tree of 207 HBc sequences from 10 genotypes of HBV reported globally.

**Table 1 microorganisms-11-02826-t001:** Position and sequence of highly conserved regions of HBx which might be used for peptide vaccine.

Position of Peptides	Sequence of Peptides
6–20	CCQLDPARDVLCLRP
48–59	DHGAHLSLRGLP
63–77	FSSAGPCALRFTSAR
93–101	LHKRTLGLS
132–143	FVLGGCRHKLVC
145–154	PAPCNFFTSA

**Table 2 microorganisms-11-02826-t002:** Position and sequence of highly conserved regions of HBc which might be used for peptide vaccine.

Position of Peptides	Sequence of Peptides
3–11	IDPYKEFGA
13–26	VELLSFLPSDFFPS
29–39	DLLDTASALYR
42–66	LESPEHCSPHHTALRQAILCWGELM
99–115	QLLWFHISCLTFGRETV
119–128	LVSFGVWIRT
129–152	PPAYRPPNAPILSTLPETTVVRRR
157–176	RRRTPSPRRRRSQSPRRRRS

**Table 3 microorganisms-11-02826-t003:** HBx-related B-cell epitopes and their conservancy analysis from HBV genotypes A–J. * indicates highest conservancy.

Name	B-Cell Epitopes	Peptide Length	Epitope Conservancy (%)
X-B1	RLCCQLD	7	70
X-B2 *	DVLCLRP	7	100
X-B3	LRGLPVCAFS	10	70
X-B4 *	AGPCALR	7	100
X-B5	ILPKVL	6	40
X-B6	LKVFVLGG	8	60
X-B7	HKLVCSPAPC	10	60

**Table 4 microorganisms-11-02826-t004:** B-cell epitopes and their conservation in HBV HBc sequences from all ten genotypes of HBV.

Name	B-Cell Epitopes	Peptide Length	Epitope Conservancy (%)
C-B1	SVELLSFLPSD	11	60
C-B2	FPSVRDL	7	60
C-B3	TASALYRE	8	70
C-B4	PEHCSPHHTALRQAILCWG	19	50
C-B5	RDLVVSYV	8	30
C-B6	IRQLLWFHISCLTF	14	80
C-B7	VLEYLVSFGV	10	80

**Table 5 microorganisms-11-02826-t005:** HBx-related T-cell class-I MHC-specific epitopes and their conservancy analysis from all HBV genotypes. * indicates the highest conservancy.

Name	Class-I MHC-Specific T-Cell Epitopes	Peptide Length	Epitope Conservancy (%)
X-M1	QLDPARDVL	9	80
X-M2 *	SAGPCALRF	9	100
X-M3	VLHKRTLGL	9	80
X-M4	VLCLRPVGA	9	80
X-M5 *	VLGGCRHKL	9	100
X-M6	CQLDPARDV	9	80
X-M7	HLSLRGLPV	9	90
X-M8 *	FVLGGCRHK	9	100
X-M9	ALRFTSARR	9	70
X-M10	MAARLCCQL	9	70
X-M11 *	CALRFTSAR	9	100
X-M12 *	SSAGPCALR	9	100
X-M13	DHGAHLSLR	9	70
X-M14	LRFTSARRM	9	70
X-M15	DPARDVLCL	9	80
X-M16	LRGLPVCAF	9	70
X-M17	RRMETTVNA	9	70
X-M18	ARRMETTVN	9	70
X-M19	ARLCCQLDP	9	70
X-M20 *	FSSAGPCAL	9	100
X-M21	LDPARDVLC	9	80
X-M22 *	RDVLCLRPV	9	100
X-M23	GAHLSLRGL	9	80
X-M24	SARRMETTV	9	70
X-M25 *	LGGCRHKLV	9	100
X-M26	LPVCAFSSA	9	70
X-M27 *	APCNFFTSA	9	100
X-M28	FTSARRMET	9	70
X-M29	LSLRGLPVC	9	90
X-M30	SLRGLPVCA	9	70
X-M31	CLRPVGAES	9	80
X-M32 *	AGPCALRFT	9	100

**Table 6 microorganisms-11-02826-t006:** T-cell class, I MHC-specific epitopes and their conservation in HBV HBc sequences from all major HBV genotypes. * indicates the highest conservancy.

Name	Class-I MHC-Specific T-Cell Epitopes	Peptide Length	Epitope Conservancy (%)
C-M1 *	LLDTASALY	9	100
C-M2 *	DIDPYKEFG	9	100
C-M3	LPETTVVRR	9	80
C-M4 *	YLVSFGVWI	9	100
C-M5	LEYLVSFGV	9	80
C-M6 *	LLWFHISCL	9	100
C-M7 *	QLLWFHISC	9	100
C-M8 *	DLLDTASAL	9	100
C-M9 *	LVSFGVWIR	9	100
C-M10 *	HISCLTFGR	9	100
C-M11 *	AYRPPNAPI	9	100
C-M12	TLPETTVVR	9	90
C-M13 *	WIRTPPAYR	9	100
C-M14	CSPHHTALR	9	80
C-M15	TTVVRRRGR	9	70
C-M16	LKIRQLLWF	9	80
C-M17	VRRRGRSPR	9	80
C-M18	LTFGRETVL	9	80
C-M19 *	YRPPNAPIL	9	100
C-M20	MGLKIRQLL	9	80
C-M21	IRQLLWFHI	9	80
C-M22	GRETVLEYL	9	80
C-M23	RRRGRSPRR	9	80
C-M24 *	RRRTPSPRR	9	100
C-M25 *	RRRRSQSPR	9	100
C-M26 *	RRRSQSPRR	9	100
C-M27	FGRETVLEY	9	80
C-M28	RETVLEYLV	9	80
C-M29	EHCSPHHTA	9	70
C-M30 *	VELLSFLPS	9	100
C-M31	MDIDPYKEF	9	90
C-M32	QAILCWGEL	9	80
C-M33	STLPETTVV	9	90
C-M34 *	WFHISCLTF	9	100
C-M35 *	LSFLPSDFF	9	100
C-M36 *	LLSFLPSDF	9	100
C-M37	ALRQAILCW	9	80
C-M38	GLKIRQLLW	9	80
C-M39	HCSPHHTAL	9	80
C-M40 *	SPRRRRSQS	9	100
C-M41	HTALRQAIL	9	80
C-M42	NMGLKIRQL	9	80
C-M43 *	IDPYKEFGA	9	100
C-M44 *	FLPSDFFPS	9	100
C-M45 *	TPPAYRPPN	9	100

**Table 7 microorganisms-11-02826-t007:** T-cell class-II MHC-specific epitopes and their conservation in HBV HBx protein sequences from genotypes A–J. * indicates the highest conservancy.

Name	Class-II MHC-Specific T-Cell Epitopes	Peptide Length	Epitope Conservancy (%)
X-T1	VLCLRPVGA	9	80
X-T2	LRFTSARRM	9	70
X-T3	LRGLPVCAF	9	70
X-T4 *	FVLGGCRHK	9	100
X-T5	FTSARRMET	9	70
X-T6 *	VLGGCRHKL	9	100
X-T7	VLHKRTLGL	9	80
X-T8 *	FSSAGPCAL	9	100
X-T9	LCLRPVGAE	9	80
X-T10	LHKRTLGLS	9	80

**Table 8 microorganisms-11-02826-t008:** HBc-related T-cell class-II MHC-specific epitopes and their conservancy analysis in all ten genotypes of HBV. * indicates the highest conservancy.

Name	Class-II MHC-Specific T-Cell Epitopes	Peptide Length	Epitope Conservancy (%)
C-T1 *	YLVSFGVWI	9	100
C-T2 *	WFHISCLTF	9	100
C-T3 *	YRPPNAPIL	9	100
C-T4	LEYLVSFGV	9	80
C-T5	IRQLLWFHI	9	80
C-T6	VRRRGRSPR	9	80
C-T7	LKIRQLLWF	9	80
C-T8	LRQAILCWG	9	80
C-T9 *	FHISCLTFG	9	100
C-T10 *	FLPSDFFPS	9	100
C-T11	FGRETVLEY	9	80
C-T12 *	LVSFGVWIR	9	100
C-T13	VVRRRGRSP	9	80
C-T14 *	WIRTPPAYR	9	100
C-T15	MGLKIRQLL	9	80
C-T16 *	VWIRTPPAY	9	100
C-T17 *	VELLSFLPS	9	100
C-T18 *	LLWFHISCL	9	100

## Data Availability

The data are available and can be used for academic or research purposes.

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
