# Peer review of "Immunoinformatics and Evaluation of Peptide Vaccines Derived from Global Hepatitis B Viral HBx and HBc Proteins Critical for Covalently Closed Circular DNA Integrity"

_microorganisms, 2023, doi:10.3390/microorganisms11122826_

Round 1

Reviewer 1 Report

Comments and Suggestions for Authors

In this presented manuscript” Immunoinformatics and Evaluation of Peptide Vaccines Derived from Global Hepatitis B Virus Covalently Closed Circular DNA Interacting HBx and HBc Proteins“ authors aimed to designed global consensus sequences of HBx and HBc proteins to analyze variable regions and highly conserved motifs which could offer potent target sites for the development of peptide vaccine, designing site-specific RNA interference and anti-HBV inhibitors. Though, the study is well structured. However, further studies regarding the vaccine construct are recommended to enhance the scientific impact of this in silico study:  

1- Authors need to perform cross human homology prediction.

2- Authors need to perform secondary and tertiary structure prediction of vaccine construct.

2- Authors need to perform molecular docking and dynamics simulation analysis of vaccine construct against a specific immune receptor within the host (TLR4 receptor).

Comments on the Quality of English Language
Please check English Language and structure of sentences.

Author Response

Reviewer 1:

In this presented manuscript” Immunoinformatics and Evaluation of Peptide Vaccines Derived from Global Hepatitis B Virus Covalently Closed Circular DNA Interacting HBx and HBc Proteins“ authors aimed to designed global consensus sequences of HBx and HBc proteins to analyze variable regions and highly conserved motifs which could offer potent target sites for the development of peptide vaccine, designing site-specific RNA interference and anti-HBV inhibitors. Though, the study is well structured. However, further studies regarding the vaccine construct are recommended to enhance the scientific impact of this in silico study: 

 Query 1- Authors need to perform cross human homology prediction.

Authors Response to Reviewer 1:

            Dear and Very Respectfully, Thank you very much for your esteemed concerns and kind evaluation towards improvement in the study. The cross-species homology prediction involves identifying homologous genes or proteins in multiple species, which can be useful for understanding the functional conservation of genes or proteins across evolutionary lineages. Several tools can be used to perform cross-species homology prediction such as Basic Local Alignment Search Tool (BLAST) which can identify similar sequences and provide a measure of sequence similarity. Similarly, Phylogenetic tree analysis, uses multiple sequence alignments of genes or proteins to identify evolutionary relationships. Homologous sequences will cluster together in the tree. In Our study we have performed the cross-species homology prediction using CLC software with much better representations in both sequence analysis and consequently graphical representation. For details please find the available representation in

  • Figure 1. Sequence alignment of HBV genotype specific consensus sequences of the HBx protein.
  • Figure 2. Phylogenetic tree of 237 HBV HBx sequences from all ten genotypes reported across the world and global consensus sequence is shown.
  • Figure 3. Sequence analysis of the genotypes A-J of the HBc protein and global consensus sequence is shown.
  • Figure 4. Phylogenetic tree of 207 HBc sequences from 10 genotypes of HBV reported globally.

Query 2- Authors need to perform secondary and tertiary structure prediction of vaccine construct.

Authors Response to Reviewer 1:

            Dear and Very Respectfully, Thank you very much for your kind suggestion. The current study aimed to investigate the most suitable peptide vaccine sequences, which can be evaluated while constructing plasmids in laboratory settings in vivo.

Please Note that the prediction of secondary and tertiary structure for peptide vaccines is not always necessary, but it can be beneficial in certain cases depending on the specific goals of vaccine design and development.

Epitope Identification: If the goal is to identify potential epitopes within a protein or pathogen that can be used as vaccine candidates, it may not be necessary to predict the secondary or tertiary structure. Sequence-based approaches, such as linear epitope prediction, can be sufficient for this purpose.

Antigenicity Prediction: Tools like B-cell epitope prediction algorithms can help assess the likelihood that a peptide will elicit an immune response based on its primary sequence. The antigenicity of a peptide can be predicted without detailed secondary or tertiary structural information.

T-cell Epitopes: Since we were interested in identifying T-cell epitopes for a vaccine, it's often more critical to understand MHC binding affinities and T-cell receptor interactions. Predicting the secondary and tertiary structure might not be as relevant in this context.

Query 3- Authors need to perform molecular docking and dynamics simulation analysis of vaccine construct against a specific immune receptor within the host (TLR4 receptor).

Authors Response to Reviewer 1:

            Dear and Very Respectfully, Thank you very much for your suggestion. However, molecular docking and dynamics simulation analysis of a vaccine construct against a specific immune receptor (TLR4 receptor) is not directly aligned with the aims and objectives of the completed study.

Please NOTE:  TLR4 is primarily associated with the innate immune response and is not directly related to HBV replication or the mechanisms discussed in the study. The study's findings are centered on HBV cccDNA, HBc, and HBx proteins, not TLR4. The study's findings related to epitopes and conserved residues within HBx and HBc proteins are significant in the context of developing potential vaccines and anti-HBV agents. Shifting the focus to TLR4 receptor analysis may distract from these key findings and their application. The primary focus of the existing study was to understand the mechanisms underlying HBV persistence and identify potential targets for the development of effective vaccines and anti-HBV agents. The proposed molecular docking and dynamics simulation analysis for TLR4 receptor does not align with the research goals, as it is unrelated to HBV persistence and vaccine development. Maintaining research integrity and relevance is crucial. Performing analyses that are not directly related to the study's objectives could compromise the integrity and focus of the research.

Reviewer 2 Report

Comments and Suggestions for Authors

Saeed et al. here report the immunoinformatic study to design new peptide vaccine candidates based on the consensus sequences of HBx and HBc proteins in all major HBV genotypes. I have different concerns, and I personally believe that the manuscript  must be revised taking into account the following comments:

1.     How the selected epitopes have been validated for computational vaccinology ? In their analyses, the authors do not report any other information of the B-cell epitopes, such as their antigenicity, toxicity and allergenicity profile. Moreover, selected epitopes should be further validated for their structural and relative surface accessibility. Other computational analyses, as homology modelling/molecular docking of the epitopes, could also required to proceed for further experimental validation.

2.     The interactome of the cccDNA  is a bit underestimated in the introduction and details about the viral DNA binding  proteins, HBc and HBx, reported in the results section (parag 3.1 pg. 4), should be moved in the introduction. Moreover, a picture could be also here added to highlight the different roles of the cccDNA, HBx and HBc proteins, that contribute to the HBV pathogenesis.

3.     The discussion section is a bit long and it is difficult  to correlate the consensus sequence analysis and to the B or T-cell epitope predictions. Maybe, a summirizing table can help the reader.

Minor concerns:

1. In the Abstract section the cccDNA abbreviation is missing.

2. Typo errors in punctuation are present in some parts of the ms, tipo. For example :

i) in the Intro section pg. 2 ....Since both HBx and HBc are crucial for HBV replication. And, targeting cccDNA binding viral HBc and HBx proteins could limit viral replication. We aimed to design global consensus sequences of HBx and......;

ii) in the discussion pg. 13....

Author Response

Saeed et al. here report the immunoinformatic study to design new peptide vaccine candidates based on the consensus sequences of HBx and HBc proteins in all major HBV genotypes. I have different concerns, and I personally believe that the manuscript  must be revised taking into account the following comments:

Query 1. How the selected epitopes have been validated for computational vaccinology ? In their analyses, the authors do not report any other information of the B-cell epitopes, such as their antigenicity, toxicity and allergenicity profile. Moreover, selected epitopes should be further validated for their structural and relative surface accessibility. Other computational analyses, as homology modelling/molecular docking of the epitopes, could also required to proceed for further experimental validation.

Authors Response to Reviewer 2:

            Dear and Very Respectfully, Thank you very much for your suggestion. The study primarily focused on in silico analyses to identify potential epitopes. The study aimed to provide a starting point for further research by narrowing down potential targets. The information related to the antigenicity, toxicity, and allergenicity of the selected epitopes is missing and these aspects require detailed experimental validation. The current study primarily aimed to investigate the potential for vaccine development against Hepatitis B Viral HBx and HBc proteins which have never been done before with such vast data acquisition. Structural validation often involves experimental techniques such as X-ray crystallography or NMR spectroscopy. Such validation is typically performed once potential epitopes are identified. Detailed antigenicity, toxicity, and allergenicity assessments and subsequent experimental studies which shall be investigated at later stages in our upcoming research projects.

Query 2.     The interactome of the cccDNA is a bit underestimated in the introduction and details about the viral DNA binding  proteins, HBc and HBx, reported in the results section (parag 3.1 pg. 4), should be moved in the introduction. Moreover, a picture could be also here added to highlight the different roles of the cccDNA, HBx and HBc proteins, that contribute to the HBV pathogenesis.

Authors Response to Reviewer 2:

            Dear and Very Respectfully, Thank you very much for your suggestion. The manuscript has been revised accordingly. Also, the revised Figure 1 has been inserted depicting different roles of the cccDNA, HBx and HBc proteins, that contribute to the HBV pathogenesis.

Query 3.     The discussion section is a bit long and it is difficult  to correlate the consensus sequence analysis and to the B or T-cell epitope predictions. Maybe, a summarizing table can help the reader.

Authors Response to Reviewer 2:

            Dear and Very Respectfully, Thank you very much for your suggestion. The discussion section has been concisely revised in more comprehensive manner.

Minor concerns:

  1. In the Abstract section the cccDNA abbreviation is missing.
  2. Typo errors in punctuation are present in some parts of the ms, tipo. For example

 in the Intro section pg. 2 ....Since both HBx and HBc are crucial for HBV replication. And, targeting cccDNA binding viral HBc and HBx proteins could limit viral replication. We aimed to design global consensus sequences of HBx and......;

  1. ii) in the discussion pg. 13....

Authors Response to Reviewer 2:

            Dear and Very Respectfully, Thank you very much for your kind evaluation and suggestions. The manuscript has been thoroughly revised and all aforementioned suggestions have been incorporated accordingly.